# Discovery of Synergistic Broadly Neutralizing Antibodies Targeting Non-Dominant Epitopes on SARS-CoV-2 RBD and NTD

**DOI:** 10.3390/vaccines13060592

**Published:** 2025-05-30

**Authors:** Hualong Feng, Zuowei Wang, Ling Li, Yunjian Li, Maosheng Lu, Xixian Chen, Lin Hu, Yi Sun, Ruiping Du, Rongrong Qin, Xuanyi Chen, Liwei Jiang, Teng Zuo

**Affiliations:** 1Laboratory of Immunoengineering, Institute of Health and Medical Technology, Hefei Institutes of Physical Science, Chinese Academy of Sciences, Hefei 230031, China; hualongfeng@mail.ustc.edu.cn (H.F.); zwwang@cmpt.ac.cn (Z.W.); lingli@cmpt.ac.cn (L.L.); yunjianli@mail.ustc.edu.cn (Y.L.); lu_msh@mail.ustc.edu.cn (M.L.); cxx2019@mail.ustc.edu.cn (X.C.); linhu@mail.ustc.edu.cn (L.H.); sunyi0927@mail.ustc.edu.cn (Y.S.); drp0619@mail.ustc.edu.cn (R.D.); qrr12345@mail.ustc.edu.cn (R.Q.); q22201313@stu.ahu.edu.cn (X.C.); 2Science Island Branch, Graduate School of USTC, Hefei 230026, China

**Keywords:** SARS-CoV-2, broadly neutralizing monoclonal antibodies, epitope mapping, RBD-8, NTD Site iv, antibody cocktail, bispecific antibody

## Abstract

**Background/Objectives**: Identification and characterization of broadly neutralizing monoclonal antibodies from individuals exposed to SARS-CoV-2, either by infection or vaccination, can inform the development of next-generation vaccines and antibody therapeutics with pan-SARS-CoV-2 protection. **Methods**: Through single B cell sorting and RT-PCR, monoclonal antibodies (mAbs) were isolated from a donor who experienced a BA.5 or BF.7 breakthrough infection after three doses of inactivated vaccines. Their binding and neutralizing capacities were measured with ELISA and a pseudovirus-based neutralization assay, respectively. Their epitopes were mapped by competition ELISA and site-directed mutation. **Results**: Among a total of 67 spike-specific mAbs cloned from the donor, four mAbs (KXD643, KXD652, KXD681, and KXD686) can neutralize all tested SARS-CoV-2 variants from wild-type to KP.3. Moreover, KXD643, KXD652, and KXD681 belong to a clonotype encoded by IGHV5-51 and IGKV1-13 and recognize the cryptic and conserved RBD-8 epitope on the receptor-binding domain (RBD). In contrast, KXD686 is encoded by IGHV1-69 and IGKV3-20 and targets a conserved epitope (NTD Site iv) outside the antigenic supersite (NTD Site i) of the N-terminal domain (NTD). Notably, antibody cocktails containing these two groups of mAbs can neutralize SARS-CoV-2 more potently due to synergistic effects. In addition, bispecific antibodies derived from KXD643 and KXD686 demonstrate further improved neutralizing potency compared to antibody cocktails. **Conclusions**: These four mAbs can be developed as candidates of pan-SARS-CoV-2 antibody therapeutics through further antibody engineering. On the other hand, vaccines designed to simultaneously elicit neutralizing antibodies towards RBD-8 and NTD Site iv have the potential to provide pan-SARS-CoV-2 protection.

## 1. Introduction

As the causative agent of COVID-19, SARS-CoV-2 is an enveloped, positive-sense, single-stranded RNA virus belonging to the betacoronavirus in the Coronaviridae family [1]. The RNA genome of SARS-CoV-2 is approximately 30 kb and encodes 4 structural proteins, 16 nonstructural proteins, and several other accessory proteins [2]. Structural proteins include spike (S), nucleocapsid (N), membrane (M), and envelope (E) proteins. Nonstructural proteins form the replicase complex and have essential roles in genomic replication and subgenomic transcription. The error-prone nature of RNA-dependent RNA polymerase, a component of replicase complex, has led to rapid emergence of variants of concern (VOCs), including Alpha (B.1.1.7), Beta (B.1.351), Gamma (P.1), Delta (B.1.617.2), and Omicron (B.1.1.529), which exhibit enhanced transmissibility, immune evasion, or altered pathogenicity [3]. Moreover, Omicron has further diversified into BA.5, BA.2.75, XBB, and JN.1 lineages and caused several waves of infection worldwide [4].

The SARS-CoV-2 spike on the virus’s surface mediates viral entry into host cells [5]. Similar to other class I fusion proteins, the SARS-CoV-2 spike is a metastable trimeric glycoprotein, with each monomer consisting of S1 and S2 subunits [6]. S1 contains a signal sequence (SP), N-terminal domain (NTD), receptor-binding domain (RBD), subdomain 1 (SD1), and subdomain 2 (SD2). S2 includes a fusion peptide (FP), heptad repeat 1 (HR1), central helix (CH), connector domain (CD), heptad repeat 2 (HR2), transmembrane (TM) domain, and cytoplasmic tail (CT). In the prefusion state of spike, RBD shifts between “up” and “down” conformations. The receptor binding motif (RBM) of RBD is exposed when RBD is up, while hidden when RBD is down. After binding with an angiotensin-converting enzyme 2 (ACE2) receptor, the spike undergoes a dramatic conformational change and eventually leads to membrane fusion between SARS-CoV-2 and host cells [7].

Due to its important role in invading host cells, the spike has been taken as the primary target of vaccines and antibody therapeutics. Over the past four years, studies on the human antibody response against the SARS-CoV-2 spike have been extensively published, which suggest that neutralizing antibodies primarily target epitopes on RBD and to a lesser extent on NTD [8,9]. Currently, neutralizing epitopes on RBD are categorized into eight major groups, with RBD-1 to RBD-3 on RBM, RBD-4 and RBD-5 on the outer face, RBD-6 and RBD-7 on the inner face, and RBD-8 on the opposite side of RBM [10,11]. On the other hand, NTD-directed neutralizing antibodies largely target the NTD antigenic supersite (Site i) [12,13,14]. The other neutralizing epitopes on NTD include those recognized by C1520, C1791 (NTD Site iv), and C1717 [15].

Through prolonged evolution, the latest SARS-CoV-2 variants have accumulated more than 60 mutations on the spike and become highly resistant to neutralizing antibodies induced by earlier variants [16,17,18]. However, repeated exposures to SARS-CoV-2 variants, either by infection or vaccination, can induce broadly neutralizing antibodies (bnAbs) against SARS-CoV-2 variants and even sarbecoviruses [19,20,21]. Identification and characterization of those bnAbs can inform the development of next-generation vaccines and antibody therapeutics with pan-SARS-CoV-2 protection. In this study, we isolated monoclonal antibodies (mAbs) from a donor who experienced a BA.5 or BF.7 breakthrough infection after three doses of inactivated vaccines. Among a total of 67 spike-specific mAbs cloned from the donor, four mAbs can neutralize all tested SARS-CoV-2 variants from wild-type to KP.3. Three of them belong to a clonotype encoded by IGHV5-51 and IGKV1-13 and recognize the cryptic and conserved RBD-8 epitope on the receptor-binding domain (RBD). The other one is encoded by IGHV1-69 and IGKV3-20 and targets a conserved epitope (NTD Site iv) outside the antigenic supersite (NTD Site i) of the N-terminal domain (NTD). Notably, these two groups of mAbs can neutralize SARS-CoV-2 variants more potently when they are combined as antibody cocktails or bispecific antibodies. Taken together, our study suggests that these four mAb can be developed as candidates of pan-SARS-CoV-2 antibody therapeutics through further antibody engineering. On the other hand, we propose that vaccines can be designed to simultaneously elicit neutralizing antibodies against RBD-8 and NTD Site iv to achieve pan-SARS-CoV-2 protection.

## 2. Materials and Methods

### 2.1. Ethics Statement

This study was approved by the Ethics Committee of Hefei Institutes of Physical Science, Chinese Academy of Sciences (approval number: YXLL-2023-47). All donors provided written informed consent for the collection of information, analysis of plasma and PBMCs, and publication of data generated from their samples.

### 2.2. Human Samples

Peripheral blood samples were collected. Plasma and peripheral blood mononuclear cells (PBMCs) were separated from blood by Ficoll density gradient centrifugation. Briefly, whole blood was mixed with an equal volume of PBS. The diluted blood was slowly layered onto Ficoll (5 mL) and then centrifuged at 600× *g* for 30 min at 25 °C. After centrifugation, plasma and PBMCs were collected from the upper layer and cell layer, respectively.

### 2.3. Cell Lines

HEK293T cells were from ATCC (CRL-3216). HEK293T-hACE2 were kindly provided by Prof. Ji Wang at Sun Yat-Sen University. HEK293T and HEK293T-hACE2 cells were cultured in DMEM with 10% fetal bovine serum (FBS, C04001, VivaCell, Shanghai, China) and 1% penicillin/streptomycin (pen/strep). FreeStyle 293F cells (A41249, Thermo Fisher Scientific, Shanghai, China) were cultured in SMM 293-TII Expression Medium (M293TII, Sino Biological, Beijing, China). All cells were maintained in a 37 °C incubator at 5% CO_2_.

### 2.4. Protein Expression and Purification

The genes encoding the ectodomain of the SARS-CoV-2 spike, including WT, BA.4/5, and BF.7, were constructed with a foldon trimerization motif, a His tag, a tandem strep-tag II, and a FLAG tag at the C-terminal. The genes encoding the S1, NTD, and RBD of WT were constructed with a strep-tag II at the C-terminal. The genes encoding the S2 of WT was constructed with a His tag at the C-terminal. Human ACE2 (huACE2, 1-740) was constructed with a Strep-tag II at the C-terminal. Spike trimer, S1, S2, NTD, RBD, and huACE2 were expressed in the FreeStyle 293F cells. For FreeStyle 293F cells transfection, polyetherimide (PEI) and plasmid were mixed in serum-free medium at a ratio of 3:1 by mass and incubated at room temperature for 20 min. Then, the mixture was added dropwise to cells. Four days after transfection, supernatant was collected and purified by Streptactin Agarose Resin 4FF (20495ES60, Yeasen, Shanghai, China) or Ni-NTA Agarose (30210, Qiagen, Shanghai, China).

### 2.5. Spike-Specific Single B Cell Sorting and PCR

PBMCs were incubated with 200 nM WT spike for 30 min at 4 °C. After wash, they were stained with cell-surface antibodies: CD3-BV510 (317331, BioLegend, CA, USA), CD19-PE/Cy7 (302215, BioLegend, CA, USA), CD27-APC (356409, BioLegend, CA, USA), CD38-APC/Cy7 (356615, BioLegend, CA, USA), human IgM-AF700 (314537, BioLegend, CA, USA), human IgD-perCP/Cy5.5 (348207, BioLegend, CA, USA), anti-His-AF488 (CL488-66005, Proteintech, Wuhan, China), anti-FLAG-PE (637309, BioLegend, CA, USA), and DAPI, for 30 min at 4 °C. The stained cells were washed with FACS buffer (PBS containing 2% FBS) and resuspended in 500 μL of FACS buffer. Spike-specific single B cells were gated as DAPI^−^CD3^−^CD19^+^CD27^+^IgD^−^His^+^FLAG^+^ and sorted into 96-well PCR plates containing 4 μL of lysis buffer (0.5 × PBS, 0.1 M DTT, and RNase inhibitor) per well. After reverse transcription reaction, variable regions of heavy and light chains were amplified by nested PCR and cloned into human IgG1, Igκ, and Igλ expression vectors. Plasmids for paired heavy and light chains were co-transfected into HEK293T cells or FreeStyle 293F cells with PEI as above. Antibodies were purified with Protein A magnetic beads (L00273, GenScript, Nanjing, China).

### 2.6. ELISA

The spike, S1, S2, NTD, or RBD were coated onto 96-well ELISA plates (100 ng/well) and incubated at 4 °C overnight. After blocking with PBS containing 10% FBS (200 µL/well) at 37 °C for 2 h, a HEK293T supernatant was added to the wells (100 µL/well) and incubated at 37 °C for 1 h. HRP-conjugated goat anti-human IgG antibodies (550004, Zen-bio, Chendu, China; 1:2000 dilution) were added to the wells and incubated at 37 °C for 1 h. TMB substrate (TMB-S-004, InnoReagents, Huzhou, China) was added to the wells (100 µL/well) and incubated at room temperature for 5 min. The reaction was stopped by 10% H_2_SO_4_ (50 µL/well), and the absorbance at 450 nm was measured.

### 2.7. Pseudovirus Neutralization Assay

To generate pseudoviruses, 6 µg of a psPAX2 packaging plasmid, 6 µg of a pLenti-luciferase transfer plasmid, and 4 µg of a spike-encoding plasmid were mixed with 48 µg of PEI in 1 mL of serum-free medium and incubated at 25 °C for 20 min. Then, the mixture was added dropwise to HEK293T cells at 70% confluence in a 10 cm dish. Supernatants with pseudoviruses were collected 48 h after transfection. For the neutralization assay, 3-fold serially diluted plasma (starting at 1:20), HEK293T supernatant (starting at 1:4.5), antibodies (starting at 10 μg/mL), antibody cocktails (1:1 mixture starting at 10 µg/mL), or bispecific antibodies (starting at 10 μg/mL) were mixed with pseudoviruses at 37 °C for 1 h. HEK293T-hACE2 cells (1.5 × 10^4^ per well) were added into the mixture and incubated at 37 °C for 48 h. The supernatant was discarded, and 100 µL of 1 × PBS was added, followed by a 20 µL of Bright-Lite Luciferase Assay System (DD1204-02, Vazyme Biotech, Nanjing, China). The fluorescence value at 560 nm for each well was measured. The percentages of neutralization were determined by comparing the fluorescence of each well with a virus-only control.

### 2.8. Competition ELISA

Competition ELISA was performed as previously described [21]. Purified huACE2, BD55-1205, BD55-5483, S309, CR3022, IMCAS74, 4A8, C1520, C1717, and C1791 were labeled with HRP (PK20001, ProteinTech, Wuhan, China). The WT spike was coated onto 96-well ELISA plates (50 ng/well) and incubated at 4 °C overnight. After blocking with PBS containing 10% FBS at 37 °C for 2 h, blocking mAbs were added to the wells and incubated at 37 °C for 1 h. Then, HRP-labeled, detecting mAbs were added to the wells in the presence of blocking mAbs and incubated at 37 °C for 1 h. The ratio of blocking and detecting antibodies was 100:1. After wash, the TMB substrate was added to the wells and incubated at room temperature for 30 min. The reaction was stopped by 10% H_2_SO_4_, and the absorbance at 450 nm was measured. The percentages of signal decrease caused by blocking mAbs were calculated.

### 2.9. Sequence Analysis

Variable regions of antibody heavy and light chains from single B cells were analyzed with IgBlast. Using Python 3.12, clonotypes were analyzed with the Bio, pandas, and numpy packages. Antibodies with the same V genes and more than 85% similarity in CDR3 (nucleotide sequence) for both heavy and light chains were defined as clones from the same clonotype.

### 2.10. Calculation of Combination Index

According to the Chou–Talalay method [22], the Combination Index (CI) is calculated with the following equation:$$CI=\frac{{\left(D\right)}_{1}}{{\left(D_{x}\right)}_{1}}+\frac{{\left(D\right)}_{2}}{{\left(D_{x}\right)}_{2}}$$

In the equation, (D)_1_ and (D)_2_ refer to the concentrations of drug1 and drug2 when they are used in combination to achieve a certain level of effect, while (D_x_)_1_ and (D_x_)_2_ refer to the concentrations of drug1 and drug2 when they are used individually to achieve the same level of effect. In our neutralization assays with antibody cocktails, two mAbs were mixed at a 1:1 ratio. Therefore, we calculated the CIs for the antibody cocktails with the following equation:$$CI=\frac{{\left(IC50\right)}_{cocktail}}{2{\left(IC50\right)}_{Ab1}}+\frac{{\left(IC50\right)}_{cocktail}}{2{\left(IC50\right)}_{Ab2}}$$

### 2.11. Construction and Expression of Bispecific Antibodies

Bispecific antibodies KXD-BsAb01 and KXD-BsAb02 were constructed in IgG-scFv format. Briefly, scFvs were constructed as VL-(G_4_S)_3_-VH and fused to the C terminus of the heavy chain by a GGS linker. Then, the plasmids encoding the heavy chain and light chain were co-transfected into FreeStyle 293F cells as above. Bispecific antibodies were purified with Protein A magnetic beads (L00273, GenScript, Nanjing, China).

## 3. Results

### 3.1. Initial Screening of mAbs from an Individual Who Experienced Breakthrough Infection After Three Doses of Inactivated Vaccines

In our previous study, we randomly recruited a cohort of 11 donors to study the antibody response induced by a breakthrough infection [23]. Among them, Donor 8 received three doses of inactivated vaccines in 2021 and was infected by BA.5 or BF.7 at the end of 2022 (Figure 1A). We collected peripheral blood samples from Donor 8 at 2 months (T1), 3 months (T2), and 6 months (T3) after infection (Figure 1A). We measured plasma neutralizing antibody titers against a panel of SARS-CoV-2 variants and SARS-CoV-1. The titers against earlier variants from WT to BA.2.75 are relatively higher than the titers against later variants from BQ.1 to KP.3, which is consistent with the mutation levels of those variants (Figure 1B). In addition, the titers from T1 to T3 are rather similar.

To isolate mAbs from Donor 8, we sorted WT-spike-specific single B cells and cloned their heavy and light chain variable regions into antibody-expressing vectors (Figure 1C). In the initial screening, we transfected HEK293T cells with those vectors and collected supernatant for ELISA and neutralization analysis (Figure 1D). In total, we identified 67 mAbs binding to the WT spike. Among them, 28 mAbs show neutralization against WT, BA.4/5, or BF.7. These neutralizing mAbs target either the RBD or the NTD. Notably, most RBD-targeting neutralizing mAbs exhibit stronger neutralization against WT compared to BA.4/5 and BF.7. In contrast, NTD-targeting neutralizing mAbs show minimal neutralization against WT.

### 3.2. Identification of Four bnAbs Against SARS-CoV-2

According to their sequences, the aforementioned 28 neutralizing mAbs belong to 12 clonotypes (Figure 2). We expressed those mAbs in larger scales and measured their IC50s against 13 SARS-CoV-2 variants from WT to KP.3 (Figure 2). Among RBD-targeting mAbs, Clonotype 1 contains 10 mAbs encoded by IGHV3-9 and IGLV1-51. Their neutralization is largely restricted to earlier variants from WT to BA.2.75. Clonotype 2 includes three mAbs encoded by IGHV4-34 and IGKV3-11. The best mAb in Clonotype 2, KXD644, can neutralize all tested variants except KP.3. Clonotype 3 is encoded by IGHV5-51 and IGKV1-13. All the three mAbs in this clonotype can neutralize all tested variants, although their neutralization potency is moderate or low. Compared with earlier variants, later variants, including BA.2.86, JN.1, KP.2, and KP.3, tend to be more sensitive to these three mAbs. The other RBD-targeting mAbs are classified into Clonotype 4 to 8, and they all have limited neutralizing breadth. Among NTD-targeting mAbs, Clonotype 9 contains three mAbs, whereas Clonotype 10 to 12 have only one mAb. KXD686 from Clonotype 12, which is encoded by IGHV1-69 and IGKV3-20, exhibits similar neutralizing breadth and potency as mAbs from Clonotype 3. In contrast, mAbs from Clonotype 9 to 11 can only neutralize one (BA.4/5) or two variants (BA.4/5 and BQ.1). Taken together, we identified four mAbs (KXD643, KXD652, KXD681, and KXD686) with broad neutralization against SARS-CoV-2.

### 3.3. The Four bnAbs Recognize Non-Dominant, While Conserved, Epitopes

To elucidate the structural basis for the broad neutralization of the above four bnAbs, we moved on to characterize their epitopes. We first performed competition ELISA for all the RBD-targeting neutralizing mAbs with ACE2, BD55-1205 (RBD-1), BD55-5483 (RBD-3), S309 (RBD-5), CR3022 (RBD-7), and IMCAS74 (RBD-8) as references (Figure 3A) [11,24,25,26,27]. Distinct from other mAbs, the three bnAbs (KXD643, KXD652, and KXD681) from Clonotype 3 only show strong competition with IMCAS74. Using AlphaFold 3 [28], we also predicted their structures in complex with WT RBD, which suggests that they target an epitope similar to that of IMCAS74 (Figure 3B). Further analysis with the predicted structures reveals that their epitopes share residues including W353, R355, R357, D428, K462-I468, E516, and L518 (Figure 3C). Among all tested SARS-CoV-2 variants, no variations are found on these residues, explaining the broad neutralization of the three bnAbs (Appendix A). To confirm the predicted epitope, we generated a panel of single-mutated pseudoviruses based on KP.3 (Figure 3D). Compared with KP.3, D428K and R466A are less sensitive to the neutralization of KXD652 and KXD681. In addition, K462D completely escapes the neutralization of the three bnAbs. Overall, these results indicate that the three bnAbs from Clonotype 3 broadly neutralize SARS-CoV-2 by targeting the conserved epitope RBD-8.

We also performed competition ELISA for all the NTD-targeting neutralizing mAbs with 4A8, C1520, C1717, and C1791 as references (Figure 3E) [12,15]. As reported, 4A8 recognizes the antigenic supersite on NTD (NTD Site i), while C1520, C1717, and C1791 (NTD Site iv) target three conserved neutralizing epitopes outside the antigenic supersite. Antibodies from Clonotype 9 to 11 display potent competition with 4A8, further confirming that NTD-targeting neutralizing antibodies primarily bind to the antigenic supersite. In contrast, KXD686 from Clonotype 12 only exhibits strong competition with C1791. The predicted structure of KXD686 in complex with WT NTD further suggests that KXD686 and C1791 recognize a similar epitope (Figure 3F) [15]. In addition, the epitope of KXD686 consists of Q23, P26, Y28, F65, D80, P82-D88, K113, I235-R237, and V267, which are also highly conserved among all tested SARS-CoV-2 variants (Figure 3G and Appendix A). To confirm the predicted epitope, we constructed a panel of single-mutated pseudoviruses based on KP.3 (Figure 3H). Compared with KP.3, none of the mutants with single mutation show significant resistance to the neutralization of KXD686. So, we generated three additional pseudoviruses with double mutations. According to the IC50s, P82T&T236A and P85T&T236A are about 10-fold less sensitive to the neutralization of KXD686. In contrast, D88K&T236A is fully resistant to the neutralization of KXD686. Taken together, these results suggest that KXD686 broadly neutralizes SARS-CoV-2 by targeting the conserved NTD Site iv.

### 3.4. Antibody Cocktails and Bispecific Antibodies Based on the Four bnAbs Neutralize SARS-CoV-2 More Potently

It has been reported that antibodies against distinct epitopes can neutralize SARS-CoV-2 more potently and broadly due to synergistic effects [11,29,30,31]. To test whether these four bnAbs can also synergistically neutralize SARS-CoV-2, we mixed the RBD-targeting bnAbs (KXD643, KXD652, and KXD681) with the NTD-targeting bnAb (KXD686) at a 1:1 ratio as antibody cocktails and measured their neutralizing potency against a panel of SARS-CoV-2 variants (Figure 4A–D). Overall, the antibody cocktails have enhanced neutralization than individual mAbs, although the degree of enhancement is not constant for all the variants. For the cocktail of KXD643 and KXD686, the enhancement is more apparent for variants including WT, BA.4/5, BA.2.75, BQ.1, XBB.1.5, EG.5.1, BA.2.86, and KP.3, as the IC50s of the antibody cocktail are more than 2-fold lower than the IC50s of individual mAbs against those variants (Figure 4A,D). In contrast, the cocktail of KXD652 and KXD686 have more apparent enhancement against WT, BQ.1, and BA.2.86 (Figure 4B,D), while the cocktail of KXD681 and KXD686 have more apparent enhancement against WT, Delta, BA.1, BA.2.75, EG.5.1, BA.2.86, JN.1, and KP.3 (Figure 4C,D). Using the Chou–Talalay method [22], we also calculated the Combination Indexes (CIs) for the antibody cocktails, which further confirms that those RBD- and NTD-targeting bnAbs have synergistic effects (Figure 4E).

As the IC50s of the antibody cocktails are still above 0.1 μg/mL for most variants, we moved on to construct bispecific antibodies to achieve enhanced neutralizing potency. With KXD643 and KXD686 as templates, we cloned and expressed two tetravalent bispecific antibodies (KXD-BsAb01 and KXD-BsAb02) in the format of IgG1 linked with scFv (Figure 5A,B). Overall, both bispecific antibodies demonstrate more potent neutralization than the antibody cocktail, particularly against variants from WT to HK.3 (Figure 5C,D). Moreover, the IC50s of KXD-BsAb02 are below 0.1 μg/mL for all tested variants except WT, EG.5.1, and HK.3. In addition, KXD-BsAb02 largely maintains neutralization against mutants that partially or fully escape KXD643 and KXD686, suggesting its resilience to future SARS-CoV-2 variants (Figure 5E).

## 4. Discussion

The unstoppable evolution of SARS-CoV-2 highlights the urgency in developing vaccines and antibody therapeutics with pan-SARS-CoV-2 and even pan-sarbecovirus protection, which requires an in-depth understanding of bnAbs induced by infection or vaccination. In this study, we identified four bnAbs (KXD643, KXD652, KXD681, and KXD686) from an individual who experienced a BA.5 or BF.7 breakthrough infection after three doses of inactivated vaccines. Among them, KXD643, KXD652, and KXD681 are from an expanded B cell clonotype encoded by IGHV5-51 and IGKV1-13 and target the cryptic and conserved RBD-8 epitope. So far, mAbs recognizing RBD-8 have been occasionally reported, such as S2H97, BIOLS56, IMCAS74, 6D6, 7D6, X17, COVOX-45, ION-300, N-612-056, and WRAIR-2057 [11,32,33,34,35,36,37,38,39]. As a representative for this class of antibodies, S2H97 displays exceptional cross-reactivity among sarbcoviruses [32]. As the breadth of KXD643, KXD652, and KXD681 was only measured with SARS-CoV-2 variants in our study, whether they have similar breadth as S2H97 remains to be tested in future work. On the other hand, KXD686 is encoded by IGHV1-69 and IGKV3-20 and recognizes the conserved NTD Site iv outside the antigenic supersite of NTD. Currently, only a handful of mAbs targeting this epitope have been identified, including S2L20, C1791, C1596, C1623, and C1648, and their breadth is largely limited to SARS-CoV-2 since other sarbecoviruses are highly variable on this epitope [13,15]. It is worth mentioning that C1596 and C1623 belong to a clonotype encoded by IGHV1-69 and IGKV3-20 [15], which is the same as KXD686. Moreover, the CDRH3s of these three mAbs are as long as 21 amino acids, and their CDRL3s are exactly the same (Appendix A). According to the definition of public antibodies, we can take these mAbs as a new class of public antibodies.

Although the four bnAbs can neutralize all tested SARS-CoV-2 variants from WT to KP.3, their IC50s are relatively high, which limits their translational potential. One way to solve this problem is to optimize these bnAbs through further affinity maturation. Over the past four years, several potent pan-SARS-CoV-2 or pan-sarbecovirus bnAbs have been developed in this way [40,41,42,43,44,45,46]. One representative example of these studies is the identification of VIR-7229 [40]. From the memory B cells of individuals who had been exposed to SARS-CoV-2 by vaccination or infection, Rosen et al. isolated a mAb (S2V29) with potent neutralization against SARS-CoV-2 variants and cross-reactivity with a panel of sarbecoviruses. To further improve the cross-reactivity and neutralization potency of S2V29, they performed in vitro affinity maturation through a combination of the yeast-display system and machine learning. Eventually, they identified VIR-7229, which has unprecedented cross-reactivity to the entire family of sarbecoviruses and maintains potent neutralization against SARS-CoV-2 variants since 2019. Given the conservation of RBD-8 across sarbecoviruses, it is promising that a potent pan-sarbecovirus that can neutralize mAbs based on KXD643, KXD652, and KXD681 can be developed. On the other hand, bispecific antibodies have been demonstrated as a robust way to develop antibodies with broad and potent neutralization against SARS-CoV-2 variants and other sarbecoviruses [11,47,48,49,50]. In this study, we constructed bispecific antibodies based on KXD643 and KXD686, which exhibit more potent neutralization than individual mAbs and the antibody cocktail. Following this direction, we can move on to develop bispecific antibodies with higher neutralizing breadth and potency by combining these four bnAbs with bnAbs targeting other epitopes.

Compared with many other studies on broadly neutralizing antibodies against SARS-CoV-2, the limitations of this study include that lack of structures determined by X-ray or Cryo-EM, neutralizing activity against authentic viruses, and protection efficacy in animal models. Nonetheless, the highlight of this study is that bnAbs against RBD-8 and NTD Site iv can synergistically neutralize SARS-CoV-2. According to a recent study [47], a possible explanation for this effect is that mAbs against the NTD Site iv can stabilize the RBD in an “up” conformation and thus make the cryptic RBD-8 epitope exposed. On the other hand, given the conservation of these two epitopes, this finding underscores the importance of these two epitopes in developing vaccines with pan-SARS-CoV-2 protection. Interestingly, a previous study has shown that the combination of RBD and NTD can elicit potent protective immune responses against SARS-CoV-2 in rabbits and macaques [51]. Based on our findings, we propose that this vaccine strategy can be further improved through modifying RBD and NTD to redirect antibody response towards the non-dominant but conserved RBD-8 and NTD Site iv epitopes.

## 5. Conclusions

This study demonstrates that KXD643, KXD652, KXD681, and KXD686 can be developed as candidates of pan-SARS-CoV-2 antibody therapeutics through further antibody engineering. It also implicates that vaccines designed to simultaneously elicit neutralizing antibodies towards RBD-8 and NTD Site iv have the potential to provide pan-SARS-CoV-2 protection.

## Figures and Tables

**Figure 1 vaccines-13-00592-f001:**
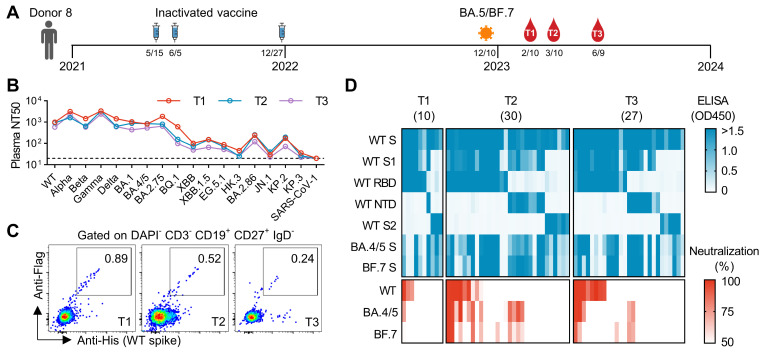
Isolation and characterization of mAbs from Donor 8. (**A**) Timeline for vaccination, infection, and sample collection. (**B**) Summary of plasma neutralizing antibody titers (NT50) against a panel of SARS-CoV-2 variants and SARS-CoV-1: NT50s are calculated with results from two independent experiments, in which duplicates are performed. (**C**) FACS plots representing the percentages of WT-spike-specific B cells at different time points. (**D**) Binding and neutralizing activities of 67 mAbs. The supernatant of HEK293T transfected with antibody-expressing plasmids was analyzed by ELISA and neutralization assay. For ELISA, the supernatant was not diluted. For neutralization, the supernatant was diluted by 4.5-fold. The results are mean values of two independent experiments, in which duplicates are performed.

**Figure 2 vaccines-13-00592-f002:**
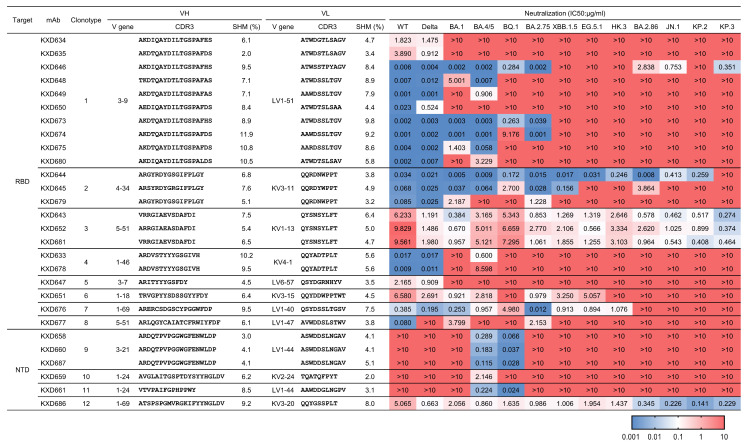
Genetic and functional characteristics of 28 neutralizing mAbs. The 28 neutralizing mAbs belong to 12 clonotypes according to sequence analysis. Their neutralizing potency and breadth are measured with 13 pseudoviruses. IC50s are calculated with results from two independent experiments, in which duplicates are performed. The highest antibody concentration used to determine IC50 is 10 μg/mL. The IC50s are color-coded, with dark blue indicating lower values and dark red indicating higher values.

**Figure 3 vaccines-13-00592-f003:**
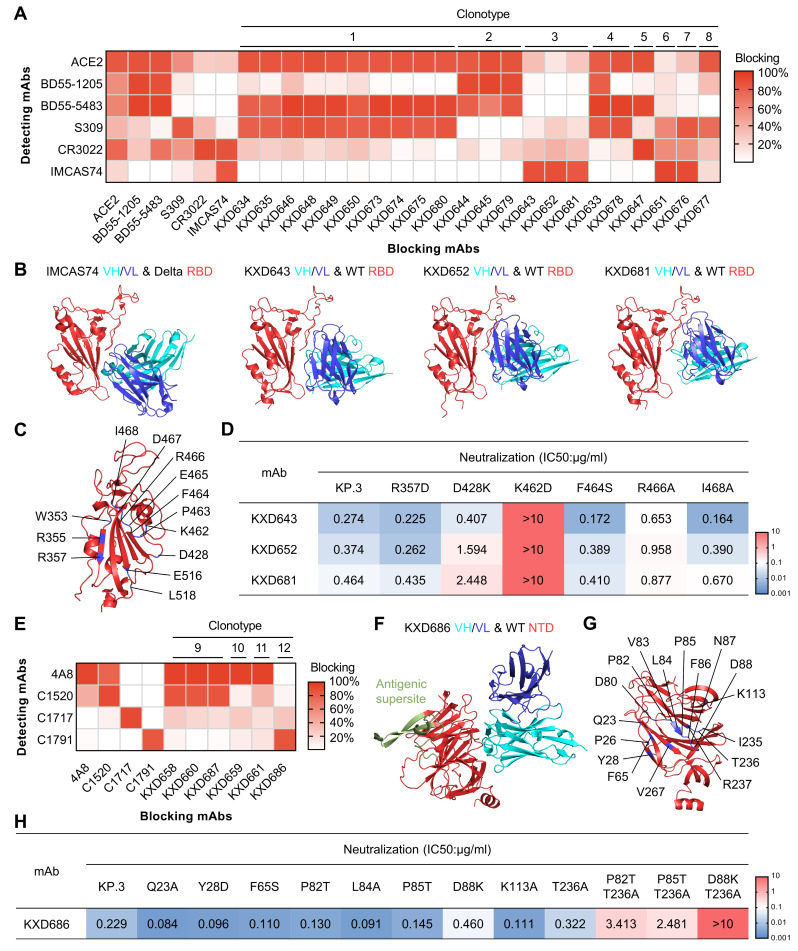
Epitope mapping and escape mutations of four bnAbs. (**A**) Competition ELISA of neutralizing mAbs against RBD: The blocking indicates signal (OD450) reduced by blocking mAbs. The results are mean values of two independent experiments, in which duplicates are performed. (**B**) The published structure of IMCAS74 in complex with Delta RBD (PDB:8HRD) and the predicted structures of KXD643, KXD652, and KXD681 in complex with WT RBD. (**C**) Common epitope of KXD643, KXD652, and KXD681 based on predicted structures. (**D**) Neutralizing activity of KXD643, KXD6523, and KXD681 against mutants based on KP.3. IC50s are calculated with results from two independent experiments, in which duplicates are performed. (**E**) Competition ELISA of neutralizing mAbs against NTD: the results are mean values of two independent experiments, in which duplicates are performed. (**F**) The predicted structure of KXD686 in complex with WT NTD: antigenic supersite on NTD is highlighted. (**G**) Epitope of KXD686 on WT NTD based on predicted structure. (**H**) Neutralizing activity of KXD686 against mutants based on KP.3. IC50s are calculated with results from two independent experiments, in which duplicates are performed.

**Figure 4 vaccines-13-00592-f004:**
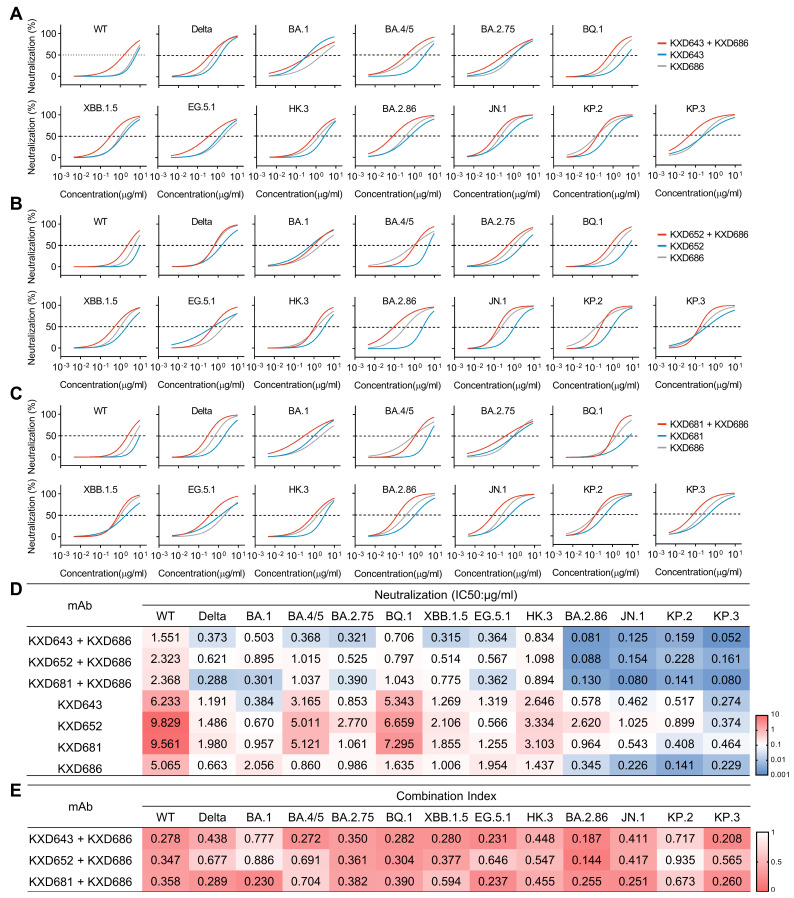
Synergistic effects of antibody cocktails. (**A**) Neutralization of KXD643 and KXD686 in individual or in a 1:1 combination: the data are represented as non-linear fit curves calculated by least squares fit. (**B**) Neutralization of KXD652 and KXD686 in individual or in a 1:1 combination: the data are represented as non-linear fit curves calculated by least squares fit. (**C**) Neutralization of KXD681 and KXD686 in individual or in a 1:1 combination: the data are represented as non-linear fit curves calculated by least squares fit. (**D**) IC50s of individual mAbs and antibody cocktails: IC50s are calculated with results from two independent experiments, in which duplicates are performed. (**E**) Summary of Combination Indexes (CIs), which are calculated with IC50s of individual mAbs and antibody cocktails.

**Figure 5 vaccines-13-00592-f005:**
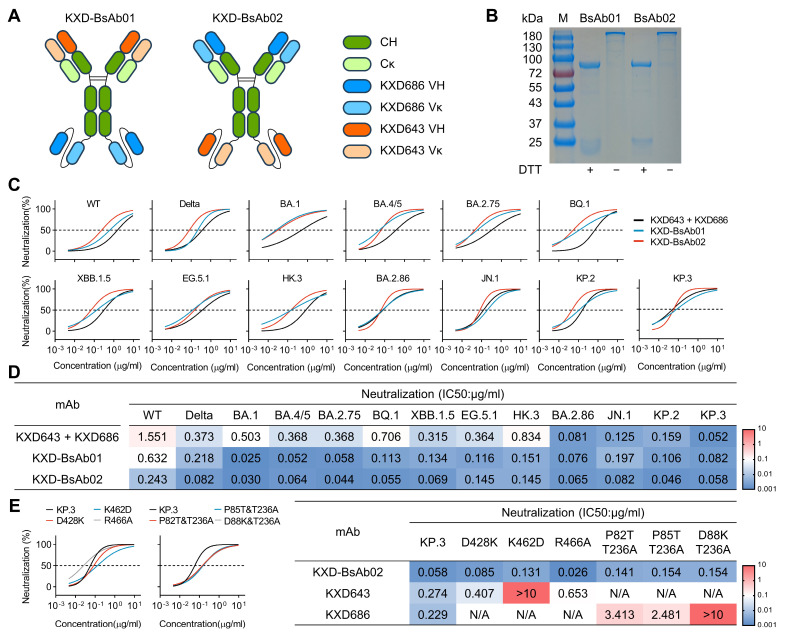
Construction and characterization of bispecific antibodies. (**A**) Schematic diagrams of two bispecific antibodies based on KXD643 and KXD686. (**B**) SDS-PAGE analysis of KXD-BsAb01 and KXD-BsAb02 under reducing (with DTT) and non-reducing conditions (without DTT): With DTT, the bispecific antibodies are disassociated. The upper bands between 72 and 100 kDa indicate the heavy chains, which are fused with a scFv. The lower bands close to 25kDa indicate the light chains. Without DTT, the bispecific antibodies maintain as complexes (the band above 180 kDa). (**C**) Neutralization of KXD-BsAb01, KXD-BsAb02, and the antibody cocktail: the data are represented as non-linear fit curves calculated by least squares fit. (**D**) IC50s of KXD-BsAb01, KXD-BsAb02, and the antibody cocktail: IC50s are calculated with results from two independent experiments, in which duplicates are performed. (**E**) Neutralization of KXD-BsAb02 against mutants based on KP.3. IC50s are calculated with results from two independent experiments, in which duplicates are performed.

## Data Availability

Data are contained within the article.

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
