# Peer review of "Discovery of Synergistic Broadly Neutralizing Antibodies Targeting Non-Dominant Epitopes on SARS-CoV-2 RBD and NTD"

_vaccines, 2025, doi:10.3390/vaccines13060592_

Round 1

Reviewer 1 Report

Comments and Suggestions for Authors

The manuscript by Wang and colleagues describes a series of extensive experiments identifying potential approaches to developing broad-based neutralizing antibodies against SARS CoV-2. Parting from monoclonal antibodies obtained from a single individual, they identified 4 potent neutralizers (broad based), three of which were directed against one epitope in the RBD. With a series of neutralization, competition and genetic experiments they identify the specific residues involved in this neutralizing capacity. They then prepared bivalent antibodies that had enhanced neutralization capacity (as did cocktails of these antibodies).

This is a very well-constructed manuscript, and the series of experiments are logical. I only have a couple of comments to improve readability:

  1. In line 213, it should be made clearer that these are the same 28 antibodies described in the previous section.
  2. In Fig 2 there is an insufficient description of the color scheme. While after reading the results I think I understood the color scheme, it is not clear how light blue and light red relate to each other, let alone white. This figure should be reconfigured with a better color scheme indicating the gradation of neutralization.
  3. The same applies to Figs 3, 4 and 5, which follow the same color scheme. Only red and blue are clear forcing the reader to read the values (obviating the help of the color).
  4. Fig 5B (the gel) needs further explanation in the legend or results.

Reviewer 2 Report

Comments and Suggestions for Authors

In this study, broadly neutralizing monoclonal antibodies derived from individuals exposed to SARS-CoV-2 or as a result of infection and vaccination were identified and characterized. Antibodies were obtained from a donor who had a breakthrough infection with BA.5 or BF.7 after three doses of inactivated vaccine using single-cell technology. The results included four monoclonal antibodies (KXD643, KXD652, 22 KXD681 and KXD686) capable of neutralizing all tested SARS-CoV-2 variants from wild-type to KP.3 KXD643, KXD652, and KXD681 antibodies were shown to belong to the same clonotype (IGHV5-51 and IGKV1-13) and to recognize the cryptic and conserved RBD-8 epitope on the receptor-binding domain (RBD). The KXD686 antibody recognizes a conserved epitope (NTD Site iv) outside the antigenic supersite (NTD Site i) of the N-terminal domain (NTD). Furthermore, the authors showed that antibody cocktails containing these two mAb groups could successfully neutralize SARS-CoV-2 more potently through synergistic effects. Even more potent aniteloa based on bispecific antibodies (KXD643 and KXD686) were obtained. 
The introduction chapter is written in detail and clear. I would like the authors to add a description of the known epitopes of neutralizing anitelles to this chapter.
Materials and methods are recommended to be described in more detail.
The results chapter has no comments.
The discussion chapter lacks a general conclusion at the end of the chapter.

Minor revision

Line 151 – Please explain and describe how you transfect cells with polyetherimide?

Reviewer 3 Report

Comments and Suggestions for Authors

The manuscript is describing an extensive study to identify, characterize and evaluate monoclonal antibodies from a donor who got infected with SARS-CoV-2 and immunized with inactivated SARS-Cov-2 vaccine.

The manuscript is well written, and the results are evident of the potential use of monoclonal antibodies as combinations or as bi-specific antibodies for use to treat SARS-Cov-2 infection.

Minor comment:

Abstract: Adjust font type and size between Background/objectives section and other sections (methods, results and conclusion) of the abstract

Reviewer 4 Report

Comments and Suggestions for Authors

Reviewer comments

vaccines-3640171

The study “Breakthrough infection elicits synergistic broadly neutralizing antibodies targeting non-dominant epitopes on RBD and NTD in an individual of inactivated SARS-CoV-2 vaccine” can be accepted after major revision.

Comments:

  1. The title is unnecessary lengthy, can be shorten.
  2. Further comparative analysis and Limitations can be included in the study.
  3. Line 112, check this “HEK293T cells were from ATCC.”
  4. Line 158, “2.8 Competition ELISA” IS this new method? provide references.
  5. Lines 282, 283. “Compared with KP.3, none of the single mutants show significant resistance 282 to the neutralization of KXD686.” Check this sentence for clarity.
  6. Detail mechanistic insights for Antibody interactions will enhance the work.
  7. Bispecific of bi-specific? Should be uniform throughout the paper.
  8. Lines 296-97 and many others. Change is-was.
  9. No conclusion section.

Round 2

Reviewer 4 Report

Comments and Suggestions for Authors

the authors have answered the comments, and the paper can be accepted.